# Computing high-degree polynomial gradients in memory

Tinish Bhattacharya [1] ✉, George H. Hutchinson [1], Giacomo Pedretti [2], Xia Sheng[2], Jim Ignowski [3], Thomas Van Vaerenbergh [4], Ray Beausoleil[4], John Paul Strachan [5,6] & Dmitri B. Strukov [1] ✉

Specialized function gradient computing hardware could greatly improve the performance of state-of-the-art optimization algorithms. Prior work on such hardware, performed in the context of Ising Machines and related concepts, is limited to quadratic polynomials and not scalable to commonly used higher-order functions. Here, we propose an approach for massively parallel gradient calculations of high-degree polynomials, which is conducive to efficient mixed-signal in-memory computing circuit implementations and whose area scales proportionally with the product of the number of variables and terms in the function and, most importantly, independent of its degree. Two flavors of such an approach are proposed. The first is limited to binary-variable polynomials typical in combinatorial optimization problems, while the second type is broader at the cost of a more complex periphery. To validate the former approach, we experimentally demonstrated solving a small-scale third-order Boolean satisfiability problem based on integrated metal-oxide memristor crossbar circuits, with competitive heuristics algorithm. Simulation results for larger-scale, more practical problems show orders of magnitude improvements in area, speed and energy efficiency compared to the state-of-the-art. We discuss how our work could enable even higher-performance systems after co-designing algorithms to exploit massively parallel gradient computation.

As the growing demand for computing processing power can be no longer supported by semiconductor technology scaling, more focus is now on developing application- and function-specific hardware accelerators. Computing gradients is essential across many applications[1–4], such as training the weights in modern Deep Neural Networks (DNN)[5] or in physics-inspired computing paradigms, including Ising machines (IMs)[6,7] or closely related approaches with Hopfield neural networks (HNNs)[8] and Boltzmann machines (BMs)[9]. For example, a continuous-time second-order HNN[10] consists of a recurrently connected network of pairwise symmetrically coupled graded-response neurons (Supplementary Fig. S1). Neuron states are continuously updated to seek the minimum of an associated scalar energy function, i.e., a Hamiltonian function in the context of Ising models[10]. For pair-wise couplings, the resulting energy function is quadratic, and hence IMs/HNNs have been extensively applied to solving quadratic unconstrained binary optimization (QUBO) problems[6] where the minimum of an arbitrary quadratic binary function is sought.

The above neuron dynamics effectively depend on the partial derivatives of the energy function with respect to the neuron values (Supplementary Fig. S1). Therefore, to rapidly converge, the most promising HNNs/IMs hardware implementations rely on massively

[1]Department of Electrical and Computer Engineering, University of California at Santa Barbara, Santa Barbara, CA, USA. [2]Artificial Intelligence Research Lab, Hewlett Packard Labs, Milpitas, CA, USA. [3]Artificial Intelligence Research Lab, Hewlett Packard Labs, Fort Collins, CO, USA. [4]Large Scale Integrated Photonics Lab, Hewlett Packard Labs, Milpitas, CA, USA. [5]Institute for Neuromorphic Compute Nodes (PGI-14), Peter Grunberg Institute, Forschungszentrum Juelich GmbH, Juelich, Germany. [6]Faculty of Electrical Engineering, RWTH Aachen University, Aachen, Germany. ✉e-mail: tinish@ucsb.edu; strukov@ece.ucsb.edu

parallel computations of gradients[6,11]. Such hardware currently constitutes the state-of-the-art in specialized gradient computing circuits. Especially promising are crossbar-circuit implementations based on analog memory devices[12,13], most importantly very dense memristors[14–17], due to prospects of efficient in-memory computing[18–21] and low footprint multi-bit implementations of the coupling weights. Though there have been proposals of higher-($K$)-order HNNs (see, e.g., Supplementary Fig. S2) and their variants[22–28] that rely on computing gradients of $K$-degree polynomials, an efficient hardware implementation is lacking, while the previous in-memory computing proposals do not readily extend to larger $K$. For example, a straightforward in-memory crossbar array implementation of $K$-order HNN with $N$ neurons (e.g., $N$ variables in energy function) requires ~$N^K$ coupling weights (Supplementary Fig. S2), i.e., hardly practical for larger $N$ and/or $K$.

Meanwhile, functions of higher $K$ can represent increasingly important and challenging problems. For example, polynomial unconstrained binary optimization (PUBO) problems[29] are described by $K$-degree polynomials and naturally arise in protein folding and other first-principle calculation methods[30–33] and operations research[34–36]. Notably, $K$ grows linearly with the size of the molecular system (molecular orbitals) in PUBO approaches for calculating electronic structure[33]. The well-known $K$–Boolean–satisfiability (K-SAT) problem goes from polynomial to expected exponential runtime as $K$ increases from 2 to 3[37]. While higher $K > 3$ can be mapped to the $K = 3$ cases, this requires a polynomial increase in the variables, potentially

increasing the runtime with increasing $K$. Interestingly, the original Hopfield network with quadratic ($K = 2$) energy function has been extended to much higher memory capacities by utilizing higher-order ($K > 2$) energy functionals[38]. However, the operation of such networks, as well as other artificial neural networks with high-order synapses[23,39,40], relies on efficient computations of higher-order polynomial gradients.

A main contribution of this work is the development of an paradigm for computing gradients of arbitrary-degree polynomial functions in a massively parallel fashion. The proposed paradigm enables efficient hardware that can immediately impact the above described use cases and can be more broadly applied to accelerate gradient computation of arbitrary functions when using Taylor expansion approximation.

## Parallel gradient computation

We expound upon binary-variable function ($H$) gradient computations using a high degree polynomial consisting of four variables ($x$) and four terms (i.e., monomials) of varying degree, each with unique factor $a$ (Fig. 1a). Computing the gradient of such a function requires calculations of all its (pseudo) partial derivatives $\Delta H_{xi} \equiv H_{xi} \Delta x_i$, where $H_{xi}$ is the difference quotient[41] of $H$ with respect to $x_i$ and $\Delta x_i$ is the change in the variable value (Supplementary Note 1). In our approach, pseudo-partial derivatives are decomposed into "make" and "break" components. (This terminology is inspired by stochastic algorithms used for solving SAT problems[42,43].) The make component of the pseudo derivative with

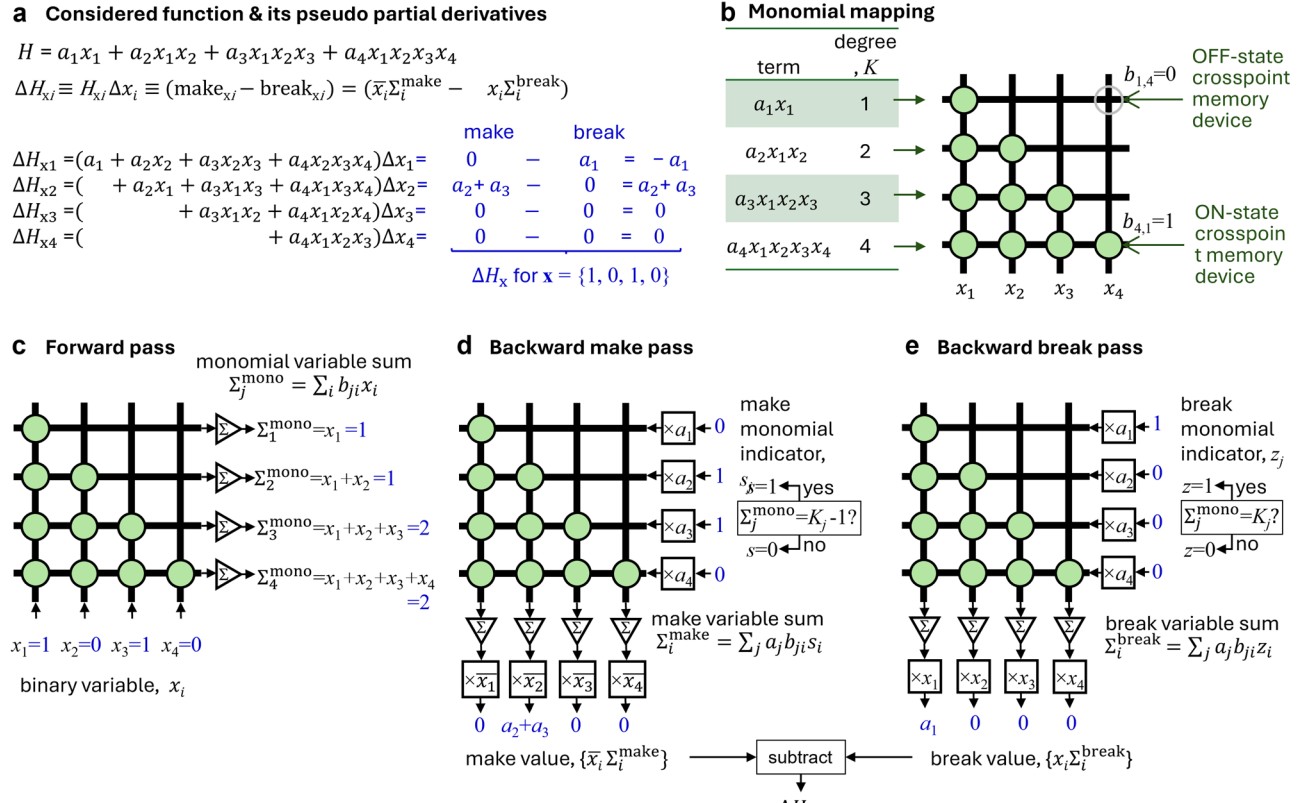

**Fig. 1 | Pseudo gradient computation for binary-variable functions.**
a Considered 4th-degree polynomial function. Note that binary-variable high-order polynomials are multilinear, i.e., the degree of any variable in a term is not more than one. **b–e** Main idea of the proposed approach showing (**b**) crossbar memory array implementation and (**c–e**) three in-memory computing operations for parallel gradient computation. **c** Sums of monomial variables are first computed in the forward vector-by-matrix multiplication pass. These values are compared to monomial orders in backward steps (**d**) and (**e**) to identify break and make type

monomials. Then, unit inputs, scaled by a monomial factor, are applied for the identified break and make monomials in the backward vector-by-matrix multiplication. Finally, the results are appropriately gated at the periphery to compute make and break terms of the pseudo-partial derivatives for each variable. The partial derivatives in question correspond to a difference between make and break components, as shown in panels (**a, d, e**). Values shown in blue correspond to specific variable assignments $x_1 = 1$, $x_2 = 0$, $x_3 = 1$, $x_4 = 0$.

respect to a given variable represents the contribution from monomials that would change their value from a zero to a non-zero one, i.e., corresponding $a$ value, after flipping the variable state. Similarly, the break component represents the contribution to the derivative from monomials that were previously evaluated to non-zero value but would become zero after changing the variable state. The difference between the make and break values is related to the pseudo partial derivative as $\Delta H_{xi} = (\text{make}_{xi} - \text{break}_{xi})$ - see an example for a specific variable assignment in Fig. 1a. We call the monomials that contribute to making and break value summations as, correspondingly, make and break monomials. Note that the sets of make and break monomials do not overlap and depend on the current variable assignment and that some monomials might be of neither make nor break type. Our goal is to design efficient hardware that can identify such make-break monomials and then compute make-and-break values for each variable.

Before discussing in-memory computing implementation, it is convenient to visualize high-order polynomials as a bipartite graph with variables and monomials representing two sets of vertices (Supplementary Fig. S3). Such a graph maps naturally to a crossbar memory array by assigning variables to one set of (say, vertical) crossbar wires while monomials to another (horizontal) set of wires by setting the array's binary coupling weight to the "on" state, i.e., $b_{ji} = 1$, if the $i$-th variable is present in a $j$-th monomial while setting the weight to the "off" state ($b_{ji} = 0$) otherwise (Fig. 1b). All pseudo derivatives are computed in parallel in three steps. First, in a "forward pass", variable values are applied to the crossbar array, and dot products $\sum_i b_{ji} x_i$ corresponding to the sums of monomial variables are calculated at the monomial side of the crossbar array (Fig. 1c). This information is then used to identify all make and break monomials (Fig. 1d, e). Specifically, monomials whose variable sum is equal to their maximum value, i.e., monomial degree $K_j$, are break monomials, while those whose sums are one short of their degree ($K_j - 1$) are made monomials. For example, for the specific variable assignment of the considered function, there are two (2nd and 3rd degree) make monomials and one (1st degree) break monomial (Fig. 1d, e).

The data flow via the crossbar circuit is reversed in the next two steps. Unit inputs for the identified monomials, denoted by indicator variable $s_j$, and zero inputs otherwise, are applied at the monomial side of the array to compute the number of monomials that each variable is a member of at the variable side of the array. In a more general case shown in Fig. 1d, in a "backward" make pass, unit inputs are scaled according to the monomial factors so that the computed dot products $\sum_j b_{ji} a_j s_j$, called make variable sums, correspond exactly to variables' potential make values. "Potential" because computed values are only relevant for currently zero variables, and only flipping those variables can change a monomial value to a non-zero one. Therefore, the make values are computed by multiplying the make variable sums at the crossbar array periphery by the inverted value of a variable, i.e., $\bar{x}_i \sum_j b_{ji} a_j s_j$, thus making the result zero for all variables that are currently one. Similar operations are performed in a "backward" break pass to compute break values, with the only difference that scaled unit inputs are applied according to the identified break monomials, denoted by indicator variable $z_j$, and the final peripheral multiplication is performed with normal variable values to compute the break values $x_i \sum_j b_{ji} a_j z_j$ (Fig. 1e). Supplementary Note 2 provides a more formal, rigorous framework for the proposed in-memory massively parallel computation of function pseudo gradient.

There are several important variations and extensions of the discussed approach. Monomial variable sums can be compared to fixed values (0 and 1) to identify make-and-break monomials by applying inverted values of variables in the forward pass (Supplementary Fig. S4a–c). This allows for simplifying peripheral circuitry of the backward passes by not requiring storing specific thresholds for each monomial of Fig. 1d, e approach. An input scaling in the backward

passes can be implemented with more complex analog (multi-level) memory devices to simplify the periphery further (Supplementary Fig. S4d). Also, backward pass computations can be implemented on separate crossbar arrays, which allows performing all three operations in parallel in a pipelined design to increase computational throughput (V).

More importantly for further discussion, the proposed approach is also suitable after minor modifications for computing pseudo gradients of Boolean logic functions expressed in conjunctive normal form (CNF), thus allowing solving Constraint Satisfaction Problems (CSPs) like SAT and MAXSAT, in native space without potentially time-consuming conversion to an equivalent PUBO form (see Supplementary Note 3 for details on such conversion). CNF Boolean function comprises conjunction (AND) of clauses, where a clause is a disjunction (OR) of literals, i.e., normal or complementary variables. The goal of the K-SAT problem, known as NP-hard for $K \geq 3$, is to find a variable assignment that satisfies all clauses of the given CNF function, with up to $K$ literals per clause. Figure 2 shows details of all steps for parallel computation of Boolean variable gain values. (A gain value is commonly used in the SAT community to describe the change in the number of satisfied clauses after flipping a variable[42,43]. It is effectively the negative of the pseudo partial derivative with respect to that variable of an equivalent PUBO energy function.) In this case, clauses are mapped to crossbar array rows, while literals (Fig. 2b), are mapped to crossbar array columns. Forward (Fig. 2c) and backward (Fig. 2d, e) operations are similar to those of the monomial approach with inverted input variables (Supplementary Fig. S4a). Supplementary Note 4 provides a more formal framework for the proposed in-memory parallel computation of CNF pseudo gradients.

Finally, another variation (Supplementary Fig. S5) is due to the use of crossbar arrays based on active, three-terminal memory devices, such as 1T1R (Supplementary Fig. S1d) or floating gate (Supplementary Fig. S1e) memory. An additional "gate" signal in such crossbar arrays (Supplementary Fig. S5a) can be used to condition the dot-product terms, which in turn allows for computing the make-and-break backward passes using a single crossbar in a single step (instead of two as shown in Supplementary Fig. S4e) for high-throughput gain computation of CNF-form Boolean functions - see Supplementary Note 5 for details.

## Experimental results

The key functionality of gain computation in CNF-form Boolean logic functions was experimentally validated by solving a high-order combinatorial optimization problem (Figs. 3, 4). Specifically, the studied optimization problem is custom-generated uniform random 3-SAT with $N = 14$ variables and $M = 64$ clauses (Supplementary Fig. S7). The problem parameters were chosen to maximize the use of hardware resources and problem hardness[44]. The WalkSAT/SKC algorithm[43], a state-of-the-art local search heuristic, was implemented to solve a 3-SAT problem. Such an algorithm repeatedly flips variables using information on their break values to converge to the solution. An unsatisfied clause is first selected randomly out of all unsatisfied clauses determined in a forward pass. Break values for all variables within the selected clause are computed in parallel in the backward pass. Then, a specific variable is flipped according to the algorithm heuristics – see Methods section for more details on the 3-SAT instance generation and algorithm.

The experiments were performed on a prototype board that features several 1T1R TaO$_x$ memristor crossbar arrays, back-end-of-the-line monolithically integrated with 180 nm CMOS circuits implementing driving, sensing, memory programming, and input/output (analog-to-digital conversion) functions (Supplementary Fig. S8). The forward and backward operations were tested on $M \times 2N$ sub-arrays of two crossbar arrays of a chip. Specifically, the conductance of on-state ($b_{ji} = 1$) memristors was first tuned to $G_{on} = 110 \, \mu S$ with 15% tuning

**a**   Considered 4SAT problem & variable gain values

$$\text{gain}_{xi} = \text{make}_{xi} - \text{break}_{xi} = \left(\bar{l}_{2i-1}\Sigma^{\text{make}}_{2i-1} + \bar{l}_{2i}\Sigma^{\text{make}}_{2i}\right) - \left(l_{2i-1}\Sigma^{\text{break}}_{2i-1} + l_{2i}\Sigma^{\text{break}}_{2i}\right)$$

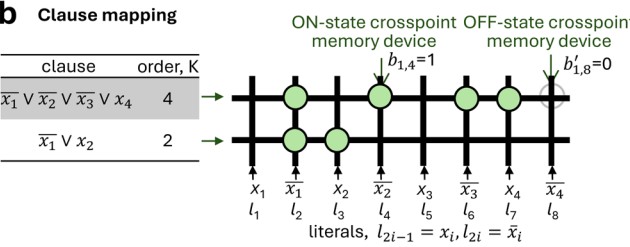

clause #1        clause #2

$$CNF = (\overline{x_1} \vee \overline{x_2} \vee \overline{x_3} \vee x_4) \wedge (\overline{x_1} \vee x_2)$$

$$H = x_1 - x_1 x_2 + x_1 x_2 x_3 - x_1 x_2 x_3 x_4$$

make    break

$$
\begin{aligned}
\text{gain}_{x1} &= 1 - x_2 + x_2 x_3 - x_2 x_3 x_4 &&= & 1 & - 0 &&= & 1 \\
\text{gain}_{x2} &= -x_1 + x_1 x_3 - x_1 x_3 x_4 &&= & 1 & - 1 &&= & 0 \\
\text{gain}_{x3} &= x_1 x_2 - x_1 x_2 x_4 &&= & 0 & - 0 &&= & 0 \\
\text{gain}_{x4} &= -x_1 x_2 x_3 &&= & 0 & - 0 &&= & 0
\end{aligned}
$$

gain values for **x** = {1, 0, 1, 0}

**b**   Clause mapping

ON-state crosspoint   OFF-state crosspoint
memory device         memory device

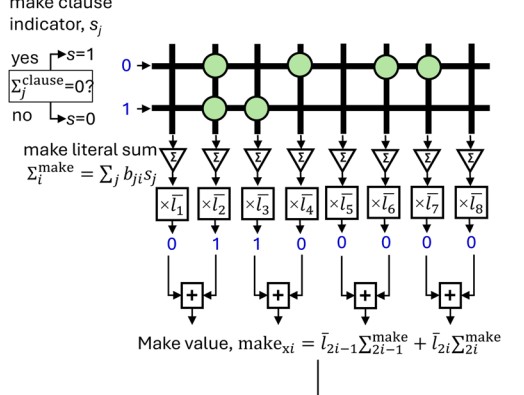

| clause | order, K |
|---|---|
| $\overline{x_1} \vee \overline{x_2} \vee \overline{x_3} \vee x_4$ | 4 |
| $\overline{x_1} \vee x_2$ | 2 |

$b_{1,4}=1$     $b'_{1,8}=0$

$x_1$ $\overline{x_1}$ $x_2$ $\overline{x_2}$ $x_3$ $\overline{x_3}$ $x_4$ $\overline{x_4}$
$l_1$ $l_2$ $l_3$ $l_4$ $l_5$ $l_6$ $l_7$ $l_8$
literals, $l_{2i-1} \equiv x_i, l_{2i} = \bar{x}_i$

**c**   Forward pass

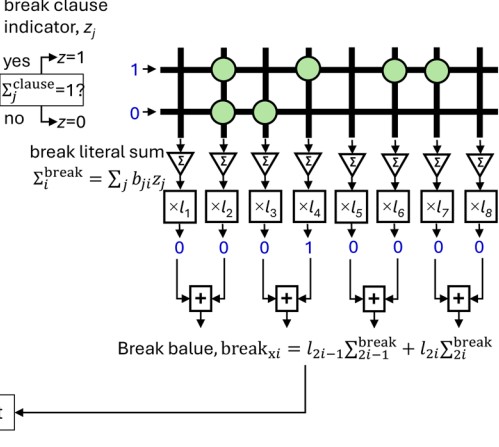

$\Sigma^{\text{clause}}_1 = \overline{x_1} + \overline{x_2} + \overline{x_3} + x_4 = 1$
$\Sigma^{\text{clause}}_2 = \overline{x_1} + x_2 = 0$

clause literal sum
$\Sigma^{\text{clause}}_j = \sum_i b_{ji} l_i$

$x_1{=}1$ $\overline{x_1}{=}0$ $x_2{=}0$ $\overline{x_2}{=}1$ $x_3{=}1$ $\overline{x_3}{=}0$ $x_4{=}0$ $\overline{x_4}{=}1$
$l_1$ $l_2$ $l_3$ $l_4$ $l_5$ $l_6$ $l_7$ $l_8$
literals, $l_{2i-1} \equiv x_i, l_{2i} \equiv \bar{x}_i$

**d**   Backward make pass

make clause
indicator, $s_j$

yes ⟶ s=1
$\Sigma^{\text{clause}}_j = 0$?
no ⟶ s=0

make literal sum
$\Sigma^{\text{make}}_i = \sum_j b_{ji} s_j$

$\times\bar{l}_1$ $\times\bar{l}_2$ $\times\bar{l}_3$ $\times\bar{l}_4$ $\times\bar{l}_5$ $\times\bar{l}_6$ $\times\bar{l}_7$ $\times\bar{l}_8$

0  1  1  0  0  0  0  0

Make value, $\text{make}_{xi} = \bar{l}_{2i-1}\Sigma^{\text{make}}_{2i-1} + \bar{l}_{2i}\Sigma^{\text{make}}_{2i}$

**e**   Backward break pass

break clause
indicator, $z_j$

yes ⟶ z=1
$\Sigma^{\text{clause}}_j = 1$?
no ⟶ z=0

break literal sum
$\Sigma^{\text{break}}_i = \sum_j b_{ji} z_j$

$\times l_1$ $\times l_2$ $\times l_3$ $\times l_4$ $\times l_5$ $\times l_6$ $\times l_7$ $\times l_8$

0  0  0  1  0  0  0  0

Break balue, $\text{break}_{xi} = l_{2i-1}\Sigma^{\text{break}}_{2i-1} + l_{2i}\Sigma^{\text{break}}_{2i}$

subtract

gain$_x$

**Fig. 2 | Conjunctive normal form gain computation. a** Considered 4th-degree CNF Boolean function, i.e., 4-SAT problem. **b–e** The main idea of the gain computing shows (**b**) crossbar memory array implementation and (**c–e**) three in-memory computing operations. The approach is similar to the polynomial pseudo gradient computation. Specifically, sums of clause literals are first computed similarly to the inverted monomial approach (Supplementary Fig. S4a) in the forward vector-by-matrix multiplication pass, as shown in panel (**c**). The sum values are compared to clause orders in backward steps (**d**) and (**e**) to identify break and make type clauses. Unit inputs, that can be scaled by clause factor when clauses in CNF are weighted (e.g., used in weighted SAT problems), are applied for the identified break and make clauses in the backward vector-by-matrix multiplication. The results are then properly gated at the periphery to compute make and break values for each variable. The corresponding variable gains (i.e., negative pseudo partial derivatives) are found by subtracting the outputs of two backward passes, as shown in panel (**a**). Values shown in blues correspond to $x_1 = 1$, $x_2 = 0$, $x_3 = 1$, $x_4 = 0$, i.e., the same example of assignment as in Fig. 1. Also note that the considered 4-SAT problem is equivalent to the polynomial function in Fig. 1a when assuming $a_1 = 1$, $a_2 = -1$, $a_3 = 1$, and $a_4 = -1$.

accuracy using the write-verify approach (Fig. 3c), while the conductance of all off-state ($b_{ji} = 0$) and outside of the utilized sub-array memristors was set to as small as possible (< 10 μS) values - see Supplementary Fig. S9 for memristor conductance map and histogram. A single iteration of the algorithm involves the application of digital voltages $V_{xi} \equiv x_i V_0$ and $V'_{xi} \equiv \bar{x}_i V_0$, with $V_0 = 0.2$ V, encoding literals $l_{2i-1} \equiv x_i$ and $l_{2i} \equiv \bar{x}_i$, correspondingly, to the word lines of the crossbar array for the forward computation (Figs. 3b and 2c). The bit-lines are tied to the ground so that the clause output currents $I_{cj}$ are computed in memory according to Kirchhoff's and Ohm's laws and correspond to dot-products $I_0 \sum_{i=1}^{2N} l_i b_{ji}$, where $I_0 \equiv V_0 G_{\text{on}}$ is a unit current via on-state coupling weight while assuming negligible off-state currents ($G_{\text{off}} = 0$), and $b_{ji}$ are coupling binary weights between literals $l_i$ and $j$-th clause. In the backward computation step, the word line voltages ($V_{cj}$) are applied to the second arrays' rows corresponding to the identified break clauses, while biasing other rows to zero. The bit line currents of the second array are multiplied by the corresponding literal values and normalized to the unit current $I_0$ to compute break values of variable $x_i$ (Figs. 3b and 2e), such that

$\text{break}_{xi} = (x_i I_{xi} + \bar{x}_i I'_{xi})/I0 = l_{2i-1}\sum_j b_{j,2i-1} z_j + l_{2i}\sum_j b_{j,2i} z_j$. Here $I_{xi}$ and $I'_{xi}$ denote the currents flowing in the bit lines corresponding to the normal ($l_{2i-1}$) and complimentary ($l_{2i}$) literals of variable $x_i$. Note that in the performed experiment, the bit line currents are always converted to the corresponding digital voltages using on-chip circuitry and transmitted out of the chip via a serial peripheral interface so that break clause checks and encode function (Fig. 3b) in the forward pass and heuristics for selecting a variable in the backward pass are performed on the personal computer (Fig. 3c).

The experimental demonstration was successful despite hardware nonidealities such as inaccurate tuning of on-state coupling weight conductances (Fig. 3c). While measured clause output currents (Fig. 4a) in the forward pass and break output currents (Fig. 4b) in the backward pass deviated from their ideal values, the margins between adjacent clause currents in the forward pass were sufficient to clearly distinguish break clauses – see, e.g., nonoverlapping histograms for 0, 1, and 2 clause current cases in Fig. 4a. The margins were slimmer for backward pass (Fig. 4b), though still large enough for correct operation, in part due to the stochastic nature of the convergence (Fig. 4c).

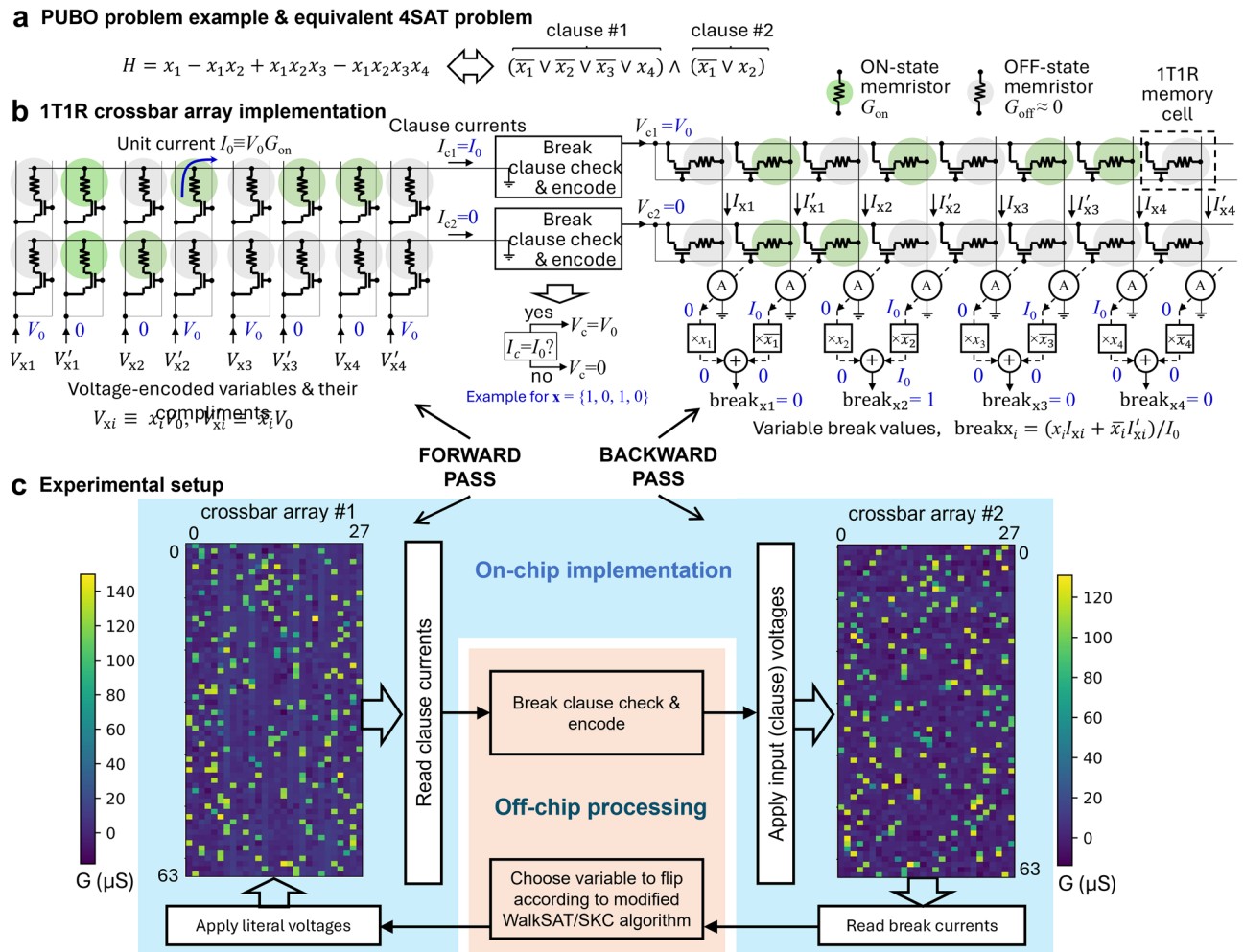

**Fig. 3 | In-memory computing hardware implementation. a** A toy example of CNF Boolean function with $M = 2$, $N = K = 4$, which is equivalent to the monomial in Fig. 1a with $a_1 = 1$, $a_2 = -1$, $a_3 = 1$, and $a_4 = -1$ when represented in a PUBO form. **b** A prospective in-memory 1T1R memristor circuit implementation of the panel (**a**) problem for parallel computation of its break values. Each 1T1R memory cell is comprised of a select transistor coupled with a memristor. Note the gate voltages are tied to the word lines in the utilized chip to suppress leakages through unselected memory cells. A text shown in blue corresponds to a specific variable assignment $x_1 = 1$, $x_2 = 0$, $x_3 = 1$, $x_4 = 0$. **c** Experimental setup details for solving the considered 14-variable 64-clause 3-SAT problem. Two arrays represent 1T1R crossbar circuits of a chip in the integrated CMOS/memristor setup and are used for demonstrating forward and backward passes. The colormaps show the measured conductance of programmed memristors corresponding to the studied 3-SAT problem in the experiment.

Notably, the run-time-distribution, i.e., the cumulative probability of finding the solution over algorithm runtime[42] in hardware, follows closely the simulated one (Fig. 4d).

## Discussion

The performance of in-memory mixed-signal computing circuits might be affected by device and circuit non-idealities, especially those of memristors. In light of these concerns, we performed detailed SPICE simulations of prospective 1T1R circuits using experimental ranges for memristor conductances and their tuning accuracies and focusing on uniform random K-SAT problems (Supplementary Note 6). We first modeled dot-product operations in forward and backward steps, specifically studying the impact of the maximum degree of K-SAT problems ($K$), clause-to-variable ratio ($M/N$) and crossbar dimension/problem size, and memristor tuning precision on the accuracy of computed clause currents, break and gain values. Simulation results show that the forward operation (Supplementary Figs. S12, S15) is more robust compared to the backward one (Supplementary Figs. S13–S17), which is explained by the simpler functionality of the former, in which only reliable detection of zero and unit clause currents is required. In backward operation, errors in break and gain values are significantly larger (Supplementary Fig. S13). However, the errors decrease with increasing $K$ due to the reduced number of clauses with $s_j$ and $z_j$ equal to one, whereas it remains relatively unaffected by increasing $M/N$ (Supplementary Fig. S14).

On the other hand, crossbar array dimensions (Supplementary Figs. S15, S16) and memristor tuning error (Supplementary Fig. S17) significantly reduce dot product accuracy and increase the overlap between neighboring adjacent break and gain values. The degradation upon crossbar array scaling is mainly due to larger unwanted leakage currents from nominally off-state memristors (with $V_0 G_{off} \sim 0.2\,\mu A$ from a single memristor). Hence, it can be addressed by improving the ON/OFF ratio of memristor technology. The alternative, more scalable solution to increasing crossbar dimensions for solving larger problems is to rely on multi-tile architecture with fixed-size smaller crossbar arrays. For example, Supplementary Fig. S24 shows one possible approach in which a logical crossbar required for mapping a large-scale combinatorial optimization problem is partitioned into multiple smaller physical crossbar arrays (tiles) that are connected with an interconnection network (Supplementary Note 12).

Crucially, the impact of studied non-idealities on the algorithmic performance is less severe, likely due to the inherent stochastic nature of heuristic algorithms and larger conductance margins in binary weight implementation as confirmed by hardware-aware modeling

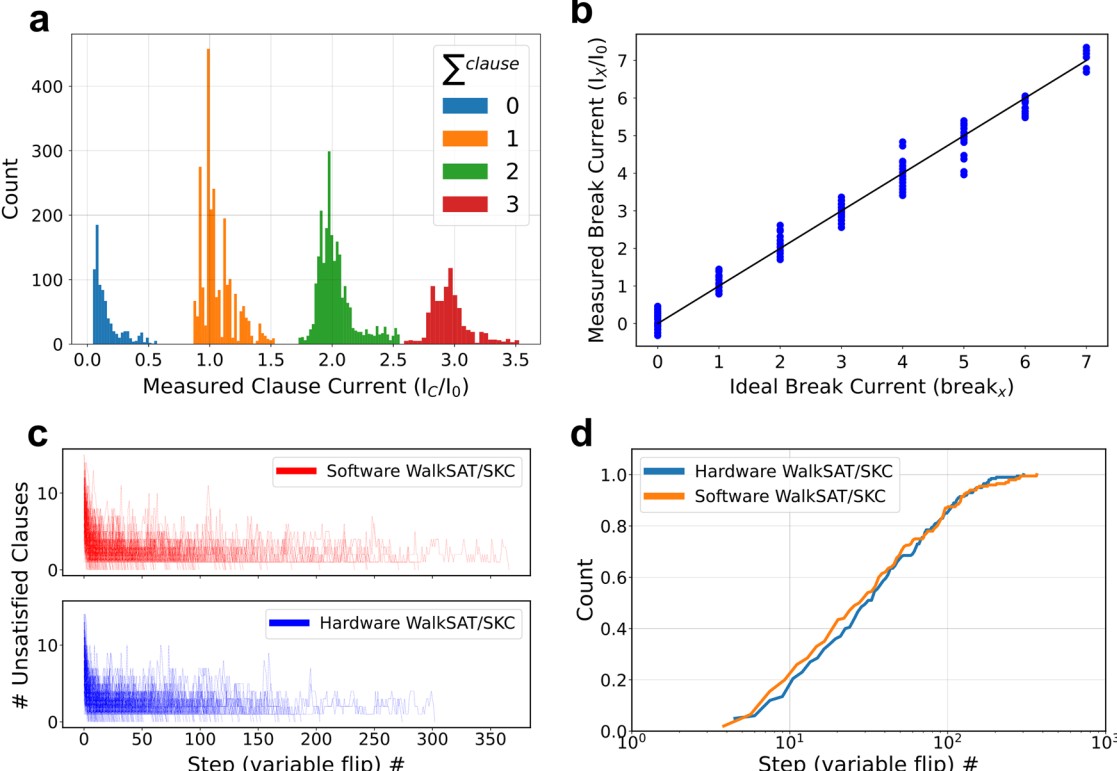

**Fig. 4 | Experimental results. a, b** Measured output (bit-line) currents for (**a**) forward and (**b**) backward passes. In panel (**a**), currents are grouped according to their ideal clause literal sum values. In panel (**b**), outputs are shown against calculated ideal break values. In both panels, data are collected during a single iteration of an algorithm. **c, d** Functional performance comparison between ideal software model and experiment, showing (**c**) evolution of the number of unsatisfied clauses, and (**d**) run-length distribution curves obtained across 200 iterations (restarts) of the algorithm, each time with new randomly initialized variable assignments. In all experiments, MAX_FLIPS = 10,000.

(Supplementary Note 9). For example, our simulations predict a negligible drop in the time-to-solution of the WalkSAT/SKC algorithm compared to the ideal (no tuning error) scenario for a 100-variable uniform random problem (Supplementary Fig. S18) assuming experimentally plausible ~2.4% and ~20% relative tuning accuracy (coefficient of variation) for high and low conductance values, respectively[45].

The proposed approach is extremely compact, requiring ~3*NM* total memory devices in the crossbar arrays, which can be used as a proxy for the overall hardware complexity, for massively parallel high-throughput computation of pseudo gradient in a polynomial with *M* monomials and *N* binary variables (Fig. 1 and Supplementary Fig. S4e). Similarly, an efficient design for computing variable gains in CNF Boolean function with *M* clauses and *N* variable features ~6*NM* memory devices (Fig. 2), which can be further reduced to ~4*NM* complexity for three-terminal memory device implementations (Supplementary Fig. S5) – see a specific example of 1T1R circuit in Supplementary Fig. S10. Figure 5 compares the crossbar area ratio between QUBO-converted problems, required for implementation with quadratic HNNs/IMs, and the proposed approach on the different SAT benchmarks[46–49] to quantify the area advantage in the context of SAT solvers. The advantage grows with the problem order exponentially, which can be analytically derived for SATs based on *K*-input XOR Boolean functions, and, e.g., ~3100 for the largest degree problem from SAT2020 competition benchmark[46] (Fig. 5 and Supplementary Fig. S11).

Moreover, Fig. 5 area advantage estimates are rather conservative. We expect a comparable area of peripheral circuitry of crossbar arrays in both quadratic HNNs/IMs and the proposed implementation. On the other hand, QUBO-converted problems feature more variables, e.g., by ~4.5× more for hard 3-SAT problems, and this coefficient grows

quickly with problem order[6]. Therefore, additional overheads are expected due to the mapping of coupling weights onto physical crossbar arrays, whose dimensions would be constrained by IR drops. Furthermore, HNNs/IMs implementations require multi-bit weights, while the proposed approach needs only binary weights, hence coming with lower programming circuitry overhead and/or enabling more compact crossbar array circuits based on conventional digital memory technologies.

Similar advantages are expected for speed and energy efficiency based on the proposed approach. Assuming negligible currents from off-state memory cells, the forward pass delay is independent of *K*. Similar to ratioed logic[50], worst-case logarithmic scaling with *K* is expected for the backward pass latency, i.e., very weak dependence of the function degree. On the system level, in the advanced process implementations, energy is largely dominated by data movement in high-performance computing circuits[51], including in-memory computing circuits[52], so energy consumption is expected to increase roughly proportional to linear circuit dimensions. Furthermore, the more compact circuitry for gradient computation could enable fitting the SAT solver completely on a chip, further improving efficiency by cutting energy and latency taxing inter-chip communication overheads. In addition, solving optimization problems in a native high-order form has led to faster convergence[22,24], partly due to additional spurious minima in the energy landscape of QUBO-converted problems[22].

For a more quantitative comparison with prior work, we developed an architecture (Supplementary Fig. S18 and Supplementary Note 7) based on the 1T1R approach and modeled physical performance by conducting hardware-aware simulations incorporating non-ideal dot-product computation due to memristor variations, timing,

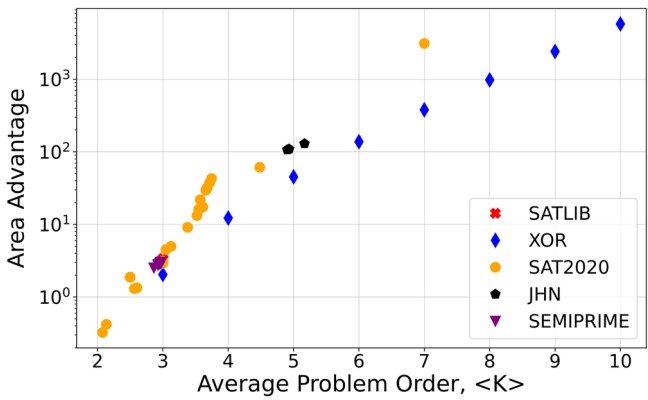

**Fig. 5 | SAT solver area advantage for the proposed approach over quadratic HNN/IM.** The area advantage is defined as the ratio of the total number of memory devices in the crossbar arrays required for both implementations. The studied benchmark problems SATLIB, XOR, SAT2020, JHN, and SEMIPRIME correspond to, respectively, uniform random 3-SAT problems[47], custom-generated $K$-input XOR problems, competition 3-SAT problems[48], "JHN" DIMACS benchmark instances from SATLIB benchmark[47], and custom-generated 3-SAT problems for semiprime factoring[49]. The higher order problems were first converted to 3-SAT with the order-reduction technique and then converted to corresponding QUBO problems using the Rosenberg approach[29]. Note that a more compact QUBO formulation can be obtained for XOR and other problems by using Tseytin transformation[63], though at the cost of substantial preprocessing overhead. Supplementary Fig. S11 shows the data used for this figure.

and energy models of crossbar array and peripheral circuits (Supplementary Tables S1, S2). Moreover, we developed a similar architecture for discrete-time binary-state high-order HNN[22] (Supplementary Fig. S19 and Supplementary Note 8) and modeled its performance using similar frameworks and assumptions (Supplementary Tables S3, S4). For simplicity, we focused on CNF-based implementation (Fig. 2, and Supplementary Fig. S10) of HNN, i.e., similar to that of WalkSAT/ SKC solver, because of CNF-based native formulation of the studied problems. The key difference of HNN architecture is gradient-descent heuristics that utilize partial derivative/gain of variables to update states (see "Methods" section), i.e., not just break values as in the case of WalkSAT/SKC solver.

Figure 6 shows the main results of the hardware performance modeling study, while Supplementary Table S5 and Supplementary Note 10 provide details of comparison with other approaches, including the Coherent Ising Machine[53], D-Wave's 2000Q quantum annealer[54], sparse Ising Machines (sIM)[55], memristor crossbar based second-order discrete-time HNNs (mem-SO-HNN)[16,22] and the Augmented Ising Machine (AIMs)[28]. WalkSAT/SKC outperforms high-order HNN in both hardware time and energy to solution metrics for larger problems. This is not surprising given the native CNF encoding of the studied problems, which is more efficiently exploited by WalkSAT/SKC heuristics. Most importantly, solvers based on our proposed approach are at least ~ 7.7 times faster than other discrete-time solvers and three orders of magnitude more energy efficient than the memristor-based second-order HNN, owing to compact hardware footprint and higher navigational efficiency of native high-order solvers. Moreover, the proposed hardware has at least two orders of magnitude higher throughput per watt and throughput per unit area compared to all other technologies.

Note that a direct comparison to recent work on high-order IMs[24] is challenging because of the lack of hardware implementation details. Also, while AIMs[28] was specifically developed to solve 3-SAT problems, a crude analysis for extending it to support $K$-order problems with $M$ clauses (by assuming an implementation with $M \times N$ array of unit cells, each hosting $K-1$ $N$:1 multiplexers) reveals inferior worst-case ~ $KN^2M$ complexity scaling (as opposed to ~ $NM$ of our approach). It is also

worth noting that a forward pass described in Fig. 2c is similar to earlier work on clause evaluation with content addressable memory implementations[56–58]. Indeed, the critical operation in the content addressable memory computation is in-memory vector-by-matrix multiplication between binary weights and inputs that produce binary "match" outputs. However, our approach takes advantage of all the outputs, not just binary match values. This feature and, more importantly, backward pass are key novelties of our approach, enabling for the first time massively parallel in-memory computing of make and break values of CNF-form Boolean functions and, more generally, pseudo gradient and gradient computation in functions of binary and real-valued variables respectively.

Furthermore, the proposed approach can be extended to computing gradients of functions with real-valued variables (Supplementary Fig. S21). Let's first note that in the simplest case of multi-linear polynomial functions, a partial derivative with respect to a given variable equals a sum of monomials that such a variable is a member of, divided by the value of that variable (Supplementary Fig. S21a). Partial derivatives are computed in parallel to follow these steps. Like the binary variable function, the first step involves an $N \times M$ crossbar array with similarly configured binary memory weights to compute in-memory monomial products. Due to real-valued variables, the products are computed differently by applying logarithmically encoded variables and exponentiating the outputs (Supplementary Fig. S21b). Appropriately weighted and specific (to the considered variable) monomial terms are summed up in the second array based on the multi-bit memory devices. Finally, the outputs from the second array are divided by variable values. Naturally, the division operation requires the variable to be nonzero, and may require a preprocessing step of shifting the variable ranges. Generalization beyond multi-linear function requires analog weights in the first array, and adequately adjusting weights in the second array (Supplementary Fig. S21d, e). Supplementary Fig. S21c shows the main idea of floating gate memory device implementation of exponentiation, while Supplementary Fig. S22 shows an example of a complete in-memory computing hybrid circuit based on memristor and floating gate memory devices. Notably, exponentiation and division are implemented directly in the second array by operating a floating gate transistor in the sub-threshold regime. The circuit operation analysis shows a trade-off between the maximum degree ($K$) and bit-precision of the monomial coefficients that can be implemented with the fixed dynamic range of the polynomial's variables (Supplementary Note 11). Preliminary simulation results using circuit and device parameters from prior in-memory computing work (Supplementary Table S6) show relative real-valued gradient error well below 10%, even without circuit optimization (Supplementary Fig. S23).

In summary, we believe the proposed hardware paradigm of computing high-degree polynomials in memory shows great promise for various applications relying on optimization algorithms, especially combinatorial optimization. Its versatility and scalability could enable the implementation of more advanced and powerful discrete-time heuristics, like G2WSAT[59], for solving constraint satisfaction problems. Its capability of computing real-valued gradients could pave the way towards implementing large-scale continuous-time continuous-state dynamics-based solver hardware potentially much faster than their discrete-time counterparts. The efficient hardware for solving problems natively in the high-order space could also boost new efficient problem embeddings, such as for quadratic assignment problems, by eliminating commonly used one-hot encoding in favor of much more compact high-order problem formulations[36].

## Methods
### 3-SAT instance generation
Random uniform 3-SAT problems with $M$ clauses and $N$ variables are generated by initializing an empty list of clauses and repeatedly adding

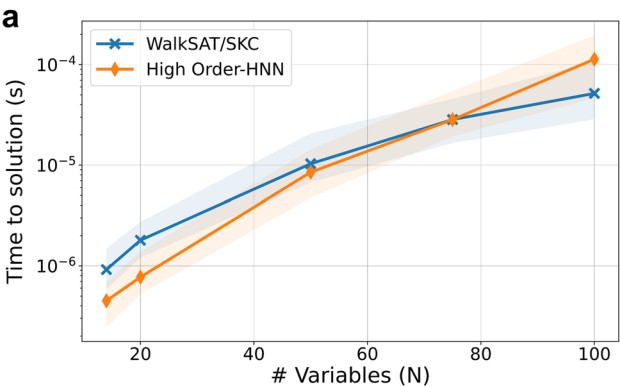

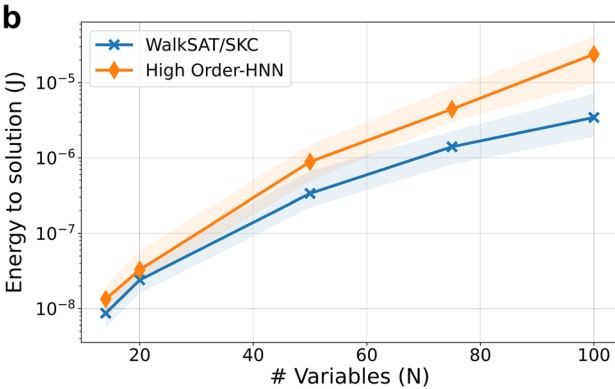

**Fig. 6 | Physical performance modeling results. a** Time and (**b**) energy to solution of the proposed WalkSAT/SKC and high-order HNN solvers for random uniform 3SAT problems. The shown data are computed based on developed hardware architecture (Supplementary Figs. S18, S19) and their models (Supplementary Table S1–S4) – see Supplementary Notes 7, 8 for more details, and hardware-aware algorithmic time-to-99% solution certainty post-processed from cumulative distributions (Supplementary Fig. S25). The shaded area in the plot shows the interquartile range (25–75%), and the markers indicate the median across instances of that problem size. Hardware-aware algorithmic simulations are taking into account errors in break/gain values due to memristor tuning errors, with the assumed relative value of ~2.4% and ~20% (corresponding standard deviation for the conductance tuning of 3 μS and 0.25 μS) for the on- and off-state memristors, respectively.

"valid" clauses to that list, one at a time, until the number of clauses reaches the target value of $M$. Specifically, a "candidate" clause is generated by randomly sampling three out of $2N$ literals without replacement. Each of the $2N$ literals has an equal probability of being chosen. Two criteria are then checked to determine if the candidate clause is valid: If both literals of a variable are present in the candidate clause; and if the candidate clause is already present in the clause list. If both criteria are false, such a clause is added to the list. Once the entire list of clauses is ready, a SAT solver is run on the generated instance to check if it is satisfiable.

3-SAT problems with $M = 64$ clauses are considered. This number matches the linear dimension of the memristor crossbar arrays and hence corresponds to the largest SAT problems in terms of the number of clauses that can be implemented with an experimental setup without employing time-consuming time-multiplexing techniques. Prior work shows[44] that satisfiable randomly generated uniform 3-SAT problems with 14 variables, corresponding to clause-to-variable ratio $M/N \approx 4.57$, are among the hardest. This was confirmed by generating and solving multiple instances of 3-SAT problems with $M = 64$ and $N$ in the range from 12 to 21.

Supplementary Fig. S7 provides a specific instance that was solved in the experimental demonstration in the common "CNF" format.

**WalkSAT/SKC algorithm.** The implemented algorithm has the following structure:

**Input:** 3-SAT CNF-formula, MAX_FLIPS, MAX_ITER, probability $p$.

**Output:** "true", if a satisfying assignment is found, "false" otherwise

1. **for** $t = 1$ to MAX_ITER
2. Randomly initialize variable assignments $\mathbf{X} = \mathbf{X_0}$
3. **for** $f = 1$ to MAX_FLIPS
4. **if X** is a solution, **return** true
5. Randomly choose an unsatisfied clause $c$
6. Calculate a set of break values (**BV**) for all member variables of $c$
7. **if** min(**BV**) = 0, flip variable with zero break value; pick randomly in a tiebreaker
8. **else**
9. with probability $p$, select & flip a variable in the clause randomly
10. with probability $1-p$, select & flip a variable with the smallest **break value**; pick at random in a tiebreaker

Note that similarly to the original version[43], known as WalkSAT/SKC, this algorithm uses only break values as deciding metric.

**Discrete-time binary-state high-order Hopfield neural network algorithm.** The implemented algorithm has the following structure:

**Input:** High-order energy function $H$ in PUBO form, MAX_FLIPS, MAX_ITER, temperature decrease rate $r$, Initial temperature $T_0$, and offset_increase_rate.

**Output:** "true", if a satisfying assignment is found, "false" otherwise.

1. **for** $t = 1$ to MAX_ITER
2. Randomly initialize variable assignments $\mathbf{X} = \mathbf{X_0}$
3. $E_{\text{offset}} = 0$
4. **for** $f = 1$ to MAX_FLIPS
5. Update temperature as $T(f) = T_0 e^{-rf}$
6. **for** each variable, $j$ **do**
7. Propose a new state as $x_j = 1$**if** $\frac{\partial H}{\partial x_j} + E_{\text{offset}}(2x_j - 1) < \eta_j$**else** 0
8. If a new state resulted in a variable flip, record
9. **if** at least one flip is accepted **then**
10. Choose one flip uniformly at random amongst them and update the state
11. $E_{\text{offset}} = 0$
12. **else**
13. $E_{offset} = E_{offset} +$ offset_increase_rate

Here, $\eta_j \sim N[0, \sqrt{2\pi}T(f)]$ is a random variable sampled from a normal distribution with zero mean and standard deviation proportional to the temperature at that step.

Note that this algorithm equivalently implements classical Hopfield neural network algorithms[26] if offset_increase_rate is set to zero (therefore fixing $E_{\text{offset}}$ to zero). In a physical implementation, a variable $x_j$ encodes a neuron state in the network, $\frac{\partial H}{\partial x_j}$ corresponds to the weighted feedback accumulated at each neuron in a particular step, while $\eta_j$ represents the noise added to the accumulated feedback. On the other hand, the inclusion of $E_{\text{offset}}$ and considering all variable flips in parallel are two aspects borrowed from the digital annealing algorithm[60] and have been shown to improve baseline high-order Hopfield neural network performance by up to an order[22].

**Definition of Time-to-Solution**

For the purposes of this paper, the term Time-to-Solution (TTS) is used to refer to Time-to-99% certainty, or the time taken for the solver/

heuristic to reach the optimal solution of the optimization problem with 99% certainty. Two types of TTS definitions, namely instance-wise TTS and batch TTS, are considered. Instance-wise TTS, as the name suggests, is the TTS of the heuristic when run on a specific instance, whereas the batch TTS refers to the median of the instance-wise TTS values across all instances belonging to that problem size.

For a heuristic (either WalkSAT/SKC or high-order HNN) that is run with a certain value of MAX_FLIPS and MAX_ITER, we first measure the Run-Length Distribution (RLD) $\hat{P}$, which is the cumulative distribution function of the number of variable flips required to find a solution (run-length) during successful iterations only[42]. It is defined as

$$\hat{P}(\text{run\_length} \leq j) = \frac{|\{t|\text{run\_length}(t) \leq j\}|}{\text{MAX\_ITER}},$$

where run_length($t$) is the run-length of the $t^{\text{th}}$ iteration that was successful, and |.| denotes the set cardinality function. We then compute the probability of success as

$$\text{success\_rate} = \frac{\text{\# of successful iterations}}{\text{MAX\_ITER}}.$$

Subsequently, the instance-wise TTS is computed using

$$\text{TTS}_i = \begin{cases} \text{MAX\_FLIPS} \times \frac{\log(1-0.99)}{\log(1-\text{success\_rate})} & \text{if success\_rate} < 0.99 \\ \hat{P}^{-1}(0.99) & \text{if success\_rate} \geq 0.99 \end{cases},$$

where $\text{TTS}_i$ is the TTS of instance $i$ and $\hat{P}^{-1}(0.99)$ is the inverse of the function $\hat{P}$ at 0.99.

## Experimental setup
The experimental setup consists of a custom chip hosting three $64 \times 64$ memristive crossbar arrays (Supplementary Fig. S8c) integrated with the custom printed circuit board (PCB) (Supplementary Fig. S8d), and custom-written firmware and Python scripts to communicate with the chip using software functions.

The Ta/TaO$_x$/Pt memristors were monolithically integrated in-house on CMOS circuits fabricated in a TSMC's 180 nm technology node (Supplementary Fig. S8a, b). The integration starts with the removal of silicon nitride and oxide passivation from the surface of the CMOS wafer with reactive ion etching, and a buffered oxide etch dip. Chromium and platinum bottom electrodes are then patterned with e-beam lithography and metal lift-off process, followed by reactive sputtered 4.5 nm tantalum oxide as the switching layer. The device stack is finalized by e-beam lithography patterning of sputtered tantalum and platinum metal as top electrodes.

The chip's CMOS subsystem implements digital control and analog sensing circuits for performing in-memory analog computations (Supplementary Fig. S8e). Each array utilizes digital-to-analog converters (DACs) to drive analog voltages (inputs) to the rows (i.e., word and gate lines) of the array. There are transimpedance amplifiers (TIAs) followed by sample-and-hold (S&H) circuits at the outputs to rapidly convert the currents to voltages, and sample them while providing virtual ground to the column (bit) lines. The sampled voltages are then multiplexed and converted to digital values using analog-to-digital converters (ADCs), each shared by 16 columns.

The PCB supplies DC analog reference signals to the chip, hosts a microcontroller, and provides a digital interface between the chip and the Python scripts running on a personal computer via serial communication. Ref. 61 provides more information on the memristor fabrication and its integration with CMOS circuits.

## Data availability
The data that support the findings in this paper are provided in the main text, Supplementary Information file and available code repository (ref. 62 and https://github.com/tinish123/imc_hdGrad/tree/v1.0.0). Additional data related to this study can be made available from the corresponding authors upon request.

## Code availability
Computer codes used to generate SAT instances, perform WalkSAT/SKC simulations, produce Fig. 6 of the main text, and perform real-valued gradient computation simulations are available online (ref. 62 and https://github.com/tinish123/imc_hdGrad/tree/v1.0.0). Additional codes related to this study can be made available from the corresponding authors upon request.

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

## Acknowledgements

This work is supported by the Defense Advanced Research Projects Agency (DARPA) under Air Force Research Laboratory (AFRL) contract no FA8650-23-3-7313.

## Author contributions

D.S. devised and supervised research. T.B. conceived the main idea of high-degree gradient computing, conducted the experiments, and performed circuit and architectural simulations. T.B. and G.H.H. performed algorithm simulations. X.S., J.I., and J.P.S. contributed to the memristor fabrication and experimental system development. T.B., G.H.H., G.P., and T.V.V. contributed to hardware modeling and performance benchmarking. T.B. and D.S. wrote the manuscript. R.B. led a collaboration effort. All authors analyzed and discussed the results.

## Competing interests

The authors declare no competing interests.
