## [Peer Review File · Nature Communications]

REVIEWER COMMENTS

Reviewer #1 (Remarks to the Author):

The authors introduce a novel approach to efficiently compute gradients of high-degree polynomials directly in memory. This method is significant, being the first in-memory hardware implementation capable of computing K-degree polynomials and achieving the best known scaling concerning the number of variables and terms/clauses in the polynomial/CNF function. Previous in-memory hardware solutions, limited to quadratic or cubic polynomials, did not scale efficiently with increasing problem sizes. In general, I find this work exciting and it makes important contributions to many relevant communities, specifically, those interested in Ising machines, optimization, AI, machine learning, and non-traditional computing.

In the abstract, the authors state the area complexity grows linearly with the number of variables and terms in the function. A more precise and clearer statement for the readers would be that the area complexity increases proportionally to the product of the number of variables and terms in the function.

The authors discuss the use of make and break values in computing full gradients, yet, it is noted that only break values are required for WalkSAT. Since the idea of computing full gradients is presented, readers may gain a better understanding of the potentials and limitations of the hardware by testing other algorithms where full gradients are required through simulation. For example, updating the spins of a Hopfield network in parallel to minimize the energy function. Do the authors see a potential for their hardware to be used with such algorithms?

When full gradients are required by other algorithms. How do the measured make currents compare to ideal make currents? How would the hardware perform for larger values of K? When do the authors expect overlap in the measured clause currents and measured break currents?

The authors show the number of steps required to perform optimization for the WalkSAT algorithm implemented on hardware. What is the required time for each step, forward pass, backward pass, and peripheral computations?

Typographical error in Fig.3. “c, e Functional performance...”. e should be changed to d.

One of the primary claims made by the authors is that the proposed method can be applied to real-valued functions. This is an exciting extension. Is it possible to simulate the real-valued gradient implementation? What is the expected precision and range for realistic hardware implementations?

Reviewer #2 (Remarks to the Author):

The manuscript presents an innovative approach for massively parallel gradient calculations of high-degree polynomials with in-memory computing memristor circuits, which can be used to enhance the performance of training the weights in neural networks, Ising machines, or Boltzmann machines. The novelty of the proposed method is the area complexity is independent of its degree and scales linearly with the number of variables and terms in the polynomials function. However, there are several areas that require further clarification and expansion in order to fully evaluate the merits of the proposed approach.

The potential impact of memristor variability with the increase in array size raises concerns regarding the accuracy of the results. The authors need to specify the on/off ratio, as the G_{off} value approximating zero does not necessarily imply a negligible effect. I recommend providing a distribution map and a statistical analysis to illustrate the extent of variability within larger array sizes and its potential effect on the accuracy and reliability of the computational results. Because both the on-off ratio and the variation of the memristors will affect the judgment of the output results as the problem size and array dimensions increase. In terms of scalability, these two points should be discussed in the article.

The question of array design in relation to scalability is currently under-addressed. The manuscript would benefit from a more detailed explanation or a design guideline on how the array can accommodate different numbers of variables and terms associated with various computational problems. A section discussing the methodology for array scaling and optimization strategies should be added to provide insight into how the memristor circuit can be effectively adapted to different computational loads. As shown in Fig. S9, the ranges of N , M , and K values vary greatly across different problems, while the array size in the hardware system is fixed. How to design the array size and map the problems is an important issue. Larger array sizes make the system more susceptible to device variability and IR drop effects. On the other hand, smaller array sizes require additional circuitry to aggregate results from multiple arrays when handling larger-scale problems, which can have a significant impact on the system's accuracy, area, power consumption, and latency performance. Therefore, the analysis of this issue is also a key content that reflects the scalability of the proposed approach.

The precision of the DAC/ADC is crucial for the implementation of logarithmic and exponential functions as shown in Fig. S10. However, the manuscript lacks a detailed analysis of the precision

requirements and how they might affect latency, power consumption, and area usage. The authors should include a more thorough evaluation of the impacts of DAC/ADC precision on the overall system performance, considering also trade-offs between precision and these system parameters.

While the manuscript highlights the area benefits of the proposed approach, there is a notable paucity of data on energy consumption and speed. I suggest that the authors include experimental data or simulations to substantiate the claims made regarding energy efficiency and processing speed. This data should be comprehensive enough to compare the proposed approach with conventional methods and to demonstrate clear advantages in terms of energy consumption and latency.

In conclusion, the manuscript has the potential to contribute significantly to the field, yet it would be greatly improved by addressing these concerns. The completion of the above recommendations would provide a stronger foundation to support the claims made and help in determining the true impact of the proposed method.

Detailed Responses to Reviewers’ Comments

The authors thank all reviewers for their valuable comments and suggestions. In response, we have revised the manuscript thoroughly and believe these changes have improved our paper substantially.

This document consists of 2 parts:

- (a) a brief list of the most significant changes and the new material in the manuscript, and
- (b) detailed responses to all referees’ comments, which also document all substantive changes made in the paper’s text, figures and figure captions.

(a) Significant changes and new material in the revised version

Main text

- Inclusion of paragraphs #4 and #5 in the “Discussion” section, discussing the SPICE simulations of crossbar arrays, results related to solver hardware design, and comparison with other approaches.
- New Figure 5 in the main text, showing hardware-time-to-solution and energy-to-solution of the WALKSAT/SKC and HO-HNN solvers on uniform random 3-SAT problems of different sizes (in response to Referee #1 and #2).
- Inclusion of two sentences, starting with “Figure S24 shows how the measured real-valued...” at the end of paragraph #6 of “Discussion” section, discussing the newly added circuit implementation and simulation results of real-valued gradient computation.
- Inclusion of three sentences, starting with “Finally, Fig. S25 shows how high-degree polynomial...” at the beginning of the last paragraph of “Discussion” section, discussing the newly added details and results related to multi-tiled architecture for the proposed idea.
- Two new methods sections: one describing the High-Order Hopfield Neural Network optimization algorithm that we implemented in simulation (in response to Referee #1) and the other describing how to calculate time-to-solution and run-length-distributions.

Supplementary Information

- New Figure S10, showing the memristor conductance maps that were programmed for the hardware experiment and distribution of the ON/OFF memristor conductance (in response to Referee #2).
- New Figures S13, S14 and S15 showing the impact of higher values of K and M/N (clause-to-variable ratio) on the accuracy of measured clause current, break and gain values, based on experimentally grounded SPICE simulations (in response to Referee #1 and in part to Referee #2).

- New Figures S16, S17 and S18 and accompanying Supplementary Note 6 showing the impact of increasing crossbar array dimension (to map larger problems) and memristor tuning error on the accuracy of measured clause current, break and gain values, based on experimentally grounded SPICE simulations (in response to Referee #2).
- New Figures S19 and S20, accompanying supplementary notes 7 and 8, and Tables S1, S2, S3 and S4 outlining circuit implementation (in 32nm technology node) of WalkSAT/SKC and HO-HNN solvers, their timing diagrams and block-wise energy, delay and area details. (in response to Referee #2 and in part Referee #1).
- Fig. S21 showing experimentally grounded simulations of time-to-solution of the WALKSAT/SKC and HO-HNN solvers on uniform random 3-SAT problems of different sizes and with varying memristor tuning errors (in response to Referee #2). Accompanying Supplementary Note 9 discussing details related to the hardware aware simulations.
- New Supplementary Note 10 and Table S5, benchmarking our proposed hardware solvers based on throughput, time and energy to solution with other alternative state-of-the-art approaches (in response to Referee #2).
- New Figures S23 and S24, Table S6 and accompanying Supplementary Note 11, describing circuit implementation of real-valued gradient computation idea, simulation results of measured versus ideal real-valued gradients and discussing trade-offs in precision, maximum order (K) and dynamic range/power consumption (in response to Referee #1 and #2).
- New Figure S25 and accompanying Supplementary Note 12, describing a FPGA-inspired architecture for implementing a multi-tiled version of the proposed approach for solving larger combinatorial optimization problems (in response to Referee #2).
- New Figure S26 showing algorithmic run-length-distribution of the WALKSAT/SKC and HO-HNN heuristics on uniform random 3-SAT problems of different sizes (in response to Referee #1 and #2).

(b) Detailed responses to reviewer’s comments

For convenience, the original comments/suggestions made by the Referees are numbered and typeset in blue, our responses are provided in black, while the yellow background highlights the changes in the revised version. Some of the similar comments are grouped together to avoid redundancy in responses.

Comment 1 (Novelty):

Reviewer #1: The authors introduce a novel approach to efficiently compute gradients of high-degree polynomials directly in memory. This method is significant, being the first in-memory hardware implementation capable of computing K -degree polynomials and achieving the best known scaling concerning the number of variables and terms/clauses in the polynomial/CNF function. Previous in-memory hardware solutions, limited to quadratic or cubic polynomials, did not scale efficiently with increasing problem sizes. In general, I find this work exciting and it

makes important contributions to many relevant communities, specifically, those interested in Ising machines, optimization, AI, machine learning, and non-traditional computing.

Reviewer #2: The manuscript presents an innovative approach for massively parallel gradient calculations of high-degree polynomials with in-memory computing memristor circuits, which can be used to enhance the performance of training the weights in neural networks, Ising machines, or Boltzmann machines. The novelty of the proposed method is the area complexity is independent of its degree and scales linearly with the number of variables and terms in the polynomials function. However, there are several areas that require further clarification and expansion in order to fully evaluate the merits of the proposed approach.

It looks like all reviewers have agreed that our manuscript includes novel and important results. We thank the reviewers for this positive and encouraging feedback. Additionally, we highly appreciate the reviewers’ suggestions for improving our manuscript. We agree that further clarification and expansion is necessary to highlight the true impact of our work and we hope that the changes and additions made to the manuscript (attached herewith) address reviewers’ comments.

Comment 2 (Dot-product error):

Reviewer #1: When full gradients are required by other algorithms. How do the measured make currents compare to ideal make currents?

In random K-SAT problems, both literals of a variable (the normal and complementary) have equal probabilities of being assigned to a clause in a mutually exclusive manner. Moreover, since make and break-currents correspond to the bit-line currents of the two literals of a variable during a backward operation, we expect similar error distribution in the real versus ideal make currents when compared to the break currents. However, we do acknowledge that analyzing error in break values alone is not enough to determine the generalizability of our hardware, as the performance of other algorithms, like the High-Order Hopfield Neural Network (HO-HNN), relies on gain/gradient computation. Therefore, in addition to break values, we have included results on measured versus ideal gain currents (obtained from SPICE simulations of the backward crossbar array), in Supplementary Note 6 and the following:

- Fig. S14 d,e,f that highlights the errors in the simulated gain current with respect to the ideal gain current for increasing values of the order of SAT problem (K).
- Fig. S15 d,e,f that highlights the errors in the simulated gain current with respect to the ideal gain current for increasing values of the clause-to-variable ratio of the SAT problem (M/N).
- Fig. S17 d,e,f that highlights the errors in the simulated gain current with respect to the ideal gain current for increasing the crossbar dimension/number of variables in the SAT problem (N).
- Fig. S18 d,e,f that highlights the errors in the simulated gain current with respect to the ideal gain current for increasing memristor tuning error.

Reviewer #1: How would the hardware perform for larger values of K ? When do the authors expect overlap in the measured clause currents and measured break currents?

We thank the reviewer for raising this question. To study the impact of increasing K on dot-product error, we performed SPICE simulations of crossbar arrays mapped with SAT problems with difference values of K and with experimentally grounded memristor tuning errors. We have compiled details of all SPICE simulation results in new supplementary note 6 and added new figures S13 and S14, that deal specifically with results related to the impact of increasing K on dot-product errors. Our results show that as K increases, the number of valid clause current levels increases. This also increases the errors at higher clause current levels, but the errors at clause current levels of zero and one are not significantly affected (see Fig. S13a). As a result, make and break clause indicator signals (s_j and z_j) that rely on detecting the zero and one clause current levels, respectively, can still be computed robustly in the forward array operation. On the other hand, the number of clauses with their make/break indicator signals equal to one decrease. This results in a reduced number of levels and errors in break and gain-values, with increasing K (see Fig. S14, and detailed explanation in Supplementary Note 6). Therefore, we believe our hardware can scale robustly with increasing values of K .

Reviewer #2: The potential impact of memristor variability with the increase in array size raises concerns regarding the accuracy of the results. The authors need to specify the on/off ratio, as the G_{off} value approximating zero does not necessarily imply a negligible effect. I recommend providing a distribution map and a statistical analysis to illustrate the extent of variability within larger array sizes and its potential effect on the accuracy and reliability of the computational results. Because both the on-off ratio and the variation of the memristors will affect the judgment of the output results as the problem size and array dimensions increase. In terms of scalability, these two points should be discussed in the article.

We thank the reviewer for raising this important point about the impact of memristor variability and array size on computational performance and agree that this aspect requires further elucidation in our manuscript.

To that end, as per the reviewer’s suggestion, we have added Fig. S10 which shows the distribution of conductance pertaining to the ON and OFF state memristors in both forward and backward arrays during the experiment, as well as the conductance map. The mean conductance of ON and OFF state memristors is $108.3\mu\text{S}$ and $3.3\mu\text{S}$, respectively, resulting in approximately $G_{\text{ON}}/G_{\text{OFF}} \sim 32.8$ dynamic range. However, we note that as described in the “Experimental Results” section of the main text, the feedback-based memristor tuning procedure was implemented with very relaxed bounds. Specifically, for G_{ON} , a bi-directional relative error bound of 15% was used and for G_{OFF} , a memristor was considered successfully tuned whenever the conductance got below $10\mu\text{S}$. Since such a strategy is sub-optimal and not representative of the capabilities of the setup, we used much more competitive memristor tuning data from other experimental works done using the same/similar setup [R1, R2] for the SPICE simulations (except for the ones where the impact of varying memristor tuning precision on performance was studied).

Furthermore, we performed SPICE simulations of 1T1R crossbar arrays mapped with different SAT problems, with experimentally grounded memristor tuning errors and realistic line resistances. We have compiled details of all SPICE simulation results in new supplementary note 6. New Fig. S16 and S17 show the impact of increasing crossbar array dimensions (required to

map larger-sized SAT problems) on clause current and break/gain value computation accuracy, respectively. New Fig. S18 shows the impact of increasing memristor tuning errors on the break and gain value computation accuracy. Finally, new Fig. 5 of the main text and Fig. S21 show the impact of the above-studied dot-product computation errors on the system-level performance for increasing problem sizes (beyond what was studied in the experiments). For the system-level evaluation, we considered two algorithms: WalkSAT/SKC and High-Order Hopfield Neural Network, and their corresponding hardware based on our proposed approach of in-memory gradient computation. Details of the hardware-aware simulations is included in new supplementary note 9.

[R1]. Jiang, M., Shan, K., He, C. & Li, C. Efficient combinatorial optimization by quantum inspired parallel annealing in analogue memristor crossbar. *Nature Communications* **14** 5927 (2023).

[R2]. Sheng, Xia, *et al.* Low-conductance and multilevel CMOS-integrated nanoscale oxide memristors. *Advanced Electronic Materials* **5.9** 1800876 (2019).

Comment 3 (Generalizability to other heuristics):

Reviewer #1: The authors discuss the use of make and break values in computing full gradients, yet, it is noted that only break values are required for WalkSAT. Since the idea of computing full gradients is presented, readers may gain a better understanding of the potentials and limitations of the hardware by testing other algorithms where full gradients are required through simulation. For example, updating the spins of a Hopfield network in parallel to minimize the energy function. Do the authors see a potential for their hardware to be used with such algorithms?

We thank the reviewer for raising this important point about the full use of gain. We had chosen the WALKSAT/SKC heuristic for its competitive performance and therefore performed simulations for only that, to be consistent with the experimental demonstration. However, we do agree that discussing other heuristics like High-Order Hopfield Neural Networks (HO-HNN) that make use of full gain, would be helpful for highlighting the true generalizability of our proposal. To that end, we have added:

- A new methods sub-section named “High-Order Hopfield Neural Network Algorithm” that describes the discrete-time algorithm that we implemented.
- Fig. S20 that describes the circuit-level block diagram for implementing a heuristic like HO-HNN while using the proposed in-memory gradient computing idea at its core. New supplementary Note 8 and Tables S3-S4 with details on the operation and energy, timing and area models of constituent circuit blocks.
- Fig. 5 of the main text, Fig. S21 and new supplementary note 9, showing hardware-aware simulations of not only the WalkSAT/SKC solver but also an HO-HNN solver on uniform random 3-SAT problems of different problem sizes. Fig. S26 shows algorithmic run-length-distribution of the WALKSAT/SKC and HO-HNN heuristics on uniform random 3-SAT problems of different sizes.

Comment 4 (Scalability to larger problems):

Reviewer #2: The question of array design in relation to scalability is currently under-addressed. The manuscript would benefit from a more detailed explanation or a design guideline on how the array can accommodate different numbers of variables and terms associated with various computational problems. A section discussing the methodology for array scaling and optimization strategies should be added to provide insight into how the memristor circuit can be effectively adapted to different computational loads. As shown in Fig. S9, the ranges of N, M, and K values vary greatly across different problems, while the array size in the hardware system is fixed. How to design the array size and map the problems is an important issue. Larger array sizes make the system more susceptible to device variability and IR drop effects. On the other hand, smaller array sizes require additional circuitry to aggregate results from multiple arrays when handling larger-scale problems, which can have a significant impact on the system's accuracy, area, power consumption, and latency performance. Therefore, the analysis of this issue is also a key content that reflects the scalability of the proposed approach.

We thank the reviewer for raising this important point about the scalability of our proposed approach to larger problems that cannot be fitted on a single crossbar array. Computation using larger crossbar arrays would be limited due to increased IR drops and leakage current contributions from OFF-state crossbar devices with activated word lines (as can be seen in new Figures S16-S17). Architectures like those proposed in [R3] can be used to map dense problems into multiple tiles, where dot-product computation is done partly in analog and partly in digital domain. However, most of the industrially/scientifically relevant large high-degree problems (including all problems considered in Fig. S12) are sparse and have limited fan-in. As a result, sparsity and fan-in-aware embedding of these problems can be performed that map the larger problem into smaller crossbar arrays such that all dot-products are executed in the analog domain, thereby not requiring any ADCs. We have added new Fig. S25 and Supplementary Note 12 that describe such an FPGA-inspired architecture comprising of in-memory computing cores present as “islands” in a “sea” of routing fabric, that can be adopted to solve larger but sparse optimization problems.

[R3] A. Shafiee, *et al.* ISAAC: A convolutional neural network accelerator with in-situ analog arithmetic in crossbars. *ACM SIGARCH Computer Architecture News* **44.3** 14-26 (2016).

Comment 5 (Real-valued gradient computation):

Reviewer #1: One of the primary claims made by the authors is that the proposed method can be applied to real-valued functions. This is an exciting extension. Is it possible to simulate the real-valued gradient implementation? What is the expected precision and range for realistic hardware implementations?

Reviewer #2: The precision of the DAC/ADC is crucial for the implementation of logarithmic and exponential functions as shown in Fig. S10. However, the manuscript lacks a detailed analysis of the precision requirements and how they might affect latency, power consumption, and area usage. The authors should include a more thorough evaluation of the impacts of DAC/ADC precision on the overall system performance, considering also trade-offs between precision and these system parameters.

We thank both reviewers for suggesting this addition and have added a new Fig. S23 that shows a potential block-level circuit implementation of the real-valued gradient computing idea. New Supplementary Note 11 discusses in detail the hardware’s steady-state operation using analytical expressions describing each block. Real-valued gradient computation simulations were performed that included behavioral modeling of circuit blocks and crossbar device variations. New Fig. S24 shows the simulation results, where the simulated real-valued gradient is plotted versus the ideal gradient value, along with its relative error. New Table S6 shows the values of the parameters used for the simulations. Our analytical modeling reveals a tradeoff between (i) the maximum order of the problem that can be implemented (K), (ii) bit-precision with which monomial coefficients can be stored in the analog-tunable floating gate cells, and (iii) the dynamic range of the polynomial’s variables (that implies increased power consumption and susceptibility to PVT variations). These are discussed in detail in Supplementary Note 11. Although we analyze the functionality of the proposed idea using steady-state analysis here, a more detailed study exploring the transient properties and effect of transistor variations would be the subject of a more circuit-focused follow-up paper.

Further, note that in the context of using such hardware for approximate computing applications like continuous-time continuous-state dynamical system solvers for high-degree optimization problems, ADCs are not explicitly required. In such a solver the partial-derivative currents in the backward array could be passed to a neuron block comprising a leaky integrator followed by nonlinear activation, whose output directly determines the trajectory of the variables. Although the variables are allowed to evolve in an analog manner within their range, when global minima are reached, they settle on either end of the range. Therefore, the final solution can be inferred by using a single comparator per variable. Low precision DACs could be used to randomly initialize the states before the solver starts, though intrinsic and extrinsic to circuit noise sources could potentially be exploited to do such initializations without requiring DACs.

Comment 6 (Latency/Energy metrics and Comparison with other approaches):

Reviewer #1: The authors show the number of steps required to perform optimization for the WalkSAT algorithm implemented on hardware. What is the required time for each step, forward pass, backward pass, and peripheral computations?

The current hardware (based on a 180 nm technology node) on which we performed the experiments, can operate at 25 MHz, therefore ~40 ns for each pass through the forward/backward array. Additional latency overheads are incurred in the serial communication that is used to convey the dot-product results of each forward/backward pass to the off-chip processor. Notably the current hardware is not optimized and is used for proof-of-concept demonstration of the gradient computing idea. However, we do lay down details on a dedicated WalkSAT/SKC and HO-HNN solver hardware that incorporates the proposed ideas and is based on circuits designed in a more competitive 32nm technology node (see new Supplementary Notes 7-8, Tables S1-S4 and Fig. S19-S20). The considered hardware design is projected to operate at 500 MHz, resulting in 2ns time per step.

Reviewer #2: While the manuscript highlights the area benefits of the proposed approach, there is a notable paucity of data on energy consumption and speed. I suggest that the authors include

experimental data or simulations to substantiate the claims made regarding energy efficiency and processing speed. This data should be comprehensive enough to compare the proposed approach with conventional methods and to demonstrate clear advantages in terms of energy consumption and latency.

We thank the reviewer for raising this important point concerning the different metrics like energy, latency of the proposed approach and agree that this aspect requires further elucidation in our manuscript. To that end, we have added:

- New Fig. S19 and S20 that describe the block-level circuit implementation of solvers running the WalkSAT/SKC and HO-HNN algorithms, respectively.
- New Supplementary Notes 7-8 and Tables S1-S4, elucidating details related to the operation of the solvers and timing, energy and area models of the constituent blocks.
- Fig. 5 b and c of the main text showing the hardware time-to-solution (in seconds) and energy-to-solution (in Joules) incurred by the two solvers in solving uniform random 3-SAT problems of varying sizes. New Supplementary Note 9 and the sub-section in methods titled “Time-to-Solution” discuss in detail the hardware-aware simulations and equation for the time-to-solution metric used, respectively.
- New Table S5 and Supplementary Note 10 that compare the time-to-solution, energy-to-solution, power, area, throughput per watt and throughput per unit area of our solver with other alternative approaches.

The crude estimates show that our solver is $\sim 7.5x$ faster than the Coherent Ising Machine (based on predicted values for CIM), and $\sim 230x$ faster than a memristor-based HNN that can only implement second-order interactions. Our solver is three orders more energy efficient than the most energy-efficient alternative, i.e., the memristor-based second-order HNN solver. Our solver also has at least two orders of magnitude higher throughput per watt and throughput per unit area compared to all other technologies.

Comment 7 (Miscellaneous):

Reviewer #1: In the abstract, the authors state the area complexity grows linearly with the number of variables and terms in the function. A more precise and clearer statement for the readers would be that the area complexity increases proportionally to the product of the number of variables and terms in the function.

Reviewer #1: Typographical error in Fig.3. “c, e Functional performance...”. e should be changed to d.

We thank the reviewer for the suggestions and pointing out the typos. We have now changed “grows linearly with the number of variables and terms in the function” to “scales proportionally with the product of the number of variables and terms in the function”. We have also resolved typographical errors in the main text.

REVIEWERS' COMMENTS

Reviewer #1 (Remarks to the Author):

The authors have addressed all major concerns and I appreciate their detailed response. Thank you.

Reviewer #2 (Remarks to the Author):

It is evident that the authors have made extensive and in-depth revisions to the entire manuscript based on my comments, providing more detailed explanations and results for the proposed method. The revised manuscript now offers a more comprehensive exposition of the principles and advantages of the proposed method, showing significant performance improvements and benefits in certain specific scenarios. Additionally, simulations were added to address the scalability and generalizability, and it is expected that future innovations at the chip architecture level could bring corresponding performance gains in some areas based on the proposed method.

The only remaining issue is that Figure 4 has a noticeably different style compared to the rest of the paper, and I suggest refining it.

Overall, the innovation and superiority of the method are well demonstrated in the revised manuscript, and I therefore recommend accepting this paper.

Responses to Reviewers’ Comments

Reviewer #1 (Remarks to the Author):

The authors have addressed all major concerns and I appreciate their detailed response. Thank you.

We thank the reviewer for their time and the constructive suggestions/questions that helped improve the manuscript.

Reviewer #2 (Remarks to the Author):

It is evident that the authors have made extensive and in-depth revisions to the entire manuscript based on my comments, providing more detailed explanations and results for the proposed method. The revised manuscript now offers a more comprehensive exposition of the principles and advantages of the proposed method, showing significant performance improvements and benefits in certain specific scenarios. Additionally, simulations were added to address the scalability and generalizability, and it is expected that future innovations at the chip architecture level could bring corresponding performance gains in some areas based on the proposed method. The only remaining issue is that Figure 4 has a noticeably different style compared to the rest of the paper, and I suggest refining it.

Overall, the innovation and superiority of the method are well demonstrated in the revised manuscript, and I therefore recommend accepting this paper.

Based on the reviewer’s suggestion, we have now updated the “Area advantage vs average problem order” figure’s style (which is now Figure 5 in the final submitted version of the manuscript) to match that of the other figures.

We are glad about the reviewer’s positive assessment of the novelty and superiority of this manuscript and thank the reviewer for their time and the constructive suggestions that helped improve the manuscript.